# Bacterial Aspiration Pneumonia in Generalized Convulsive Status Epilepticus: Incidence, Associated Factors and Outcome

**DOI:** 10.3390/jcm11226673

**Published:** 2022-11-10

**Authors:** Romain Tortuyaux, Frédéric Wallet, Philippe Derambure, Saad Nseir

**Affiliations:** 1Intensive Care Unit, CHU Lille, F-59000 Lille, France; 2Department of Clinical Neurophysiology, CHU Lille, F-59000 Lille, France; 3Laboratoire de Bactériologie-Hygiène, Centre de Biologie Pathologie, CHU Lille, F-59000 Lille, France; 4CNRS, INSERM, Institut Pasteur Lille, U1019-UMR 9017-CIIL, Université de Lille, F-59000 Lille, France; 5CHU Lille, INSERM U1172, Université de Lille, F-59000 Lille, France; 6INSERM U1285, CNRS, UMR 8576-UGSF, Université de Lille, F-59000 Lille, France

**Keywords:** status epilepticus, intensive care unit, bacterial aspiration pneumonia, 3-month outcome

## Abstract

Suspicion of bacterial aspiration pneumonia (BAP) is frequent during generalized convulsive status epilepticus (GCSE). Early identification of BAP is required in order to avoid useless antibiotic therapy. In this retrospective monocentric study, we aimed to determine the incidence of aspiration syndrome and BAP in GCSE requiring mechanical ventilation (MV) and factors associated with the occurrence of BAP. Patients were older than 18 years and had GCSE requiring MV. To distinguish BAP from pneumonitis, tracheal aspirate and quantitative microbiological criterion were used. Out of 226 consecutive patients, 103 patients (46%) had an aspiration syndrome, including 54 (52%) with a BAP. *Staphylococcus aureus* represented 33% of bacterial strains. No relevant baseline characteristics differed, including serum levels of CRP, PCT, and albumin. The median duration of treatment for BAP was 7 days (5–7). Patients with BAP did not have a longer duration of MV (*p* = 0.18) and ICU stay (*p* = 0.18) than those with pneumonitis. At 3 months, 24 patients (44%) with BAP and 10 (27%) with pneumonitis had a poor functional outcome (*p* = 0.06). In conclusion, among patients with GCSE, half of the patients had an aspiration syndrome and one-quarter suffered from BAP. Clinical characteristics and biomarkers were not useful for differentiating BAP from pneumonitis. These results highlight the need for a method to rapidly differentiate BAP from pneumonitis, such as polymerase-chain-reaction-based techniques.

## 1. Introduction

Status epilepticus (SE) is broadly defined as a prolonged seizure and remains a common neurological emergency with an overall mortality approaching 20% [1]. Generalized convulsive SE is considered as the worst type of SE and may lead to neurological injury and risk of sequelae [2]. To an early seizure cessation, treatments could be aggressive and contribute to worsening consciousness, such as benzodiazepine or sedatives [3]. In addition, extra-neurological complications are frequent, especially respiratory infection, and may impact the prognosis [4,5]. Aspiration is common in patients with impaired consciousness and is probably more frequent in case of persistent convulsions [6]. Early identification of bacterial aspiration pneumonia (BAP) is needed to avoid useless treatments. In addition, BAP has been associated with acute respiratory distress syndrome and requires an early antibiotic therapy [6]. However, in the absence of a microbiological sample, it is impossible to differentiate BAP from pneumonitis. A previous study in patients with coma requiring MV did not find a relevant difference between patients, highlighting the need for early bacterial identification [7]. To our knowledge, no study has investigated the incidence of BAP and factors associated with the occurrence of BAP in patients with generalized convulsive SE requiring mechanical ventilation (MV).

We hypothesized that the incidence of aspiration syndrome and BAP would be high in a population at risk, as generalized convulsive SE patients are. Therefore, we conducted this retrospective study to determine the incidence of aspiration syndrome and BAP in order to identify factors associated with the occurrence of BAP and to study the impact of aspiration syndrome and BAP on MV duration, intensive care unit (ICU) length of stay, and 3-month outcomes.

## 2. Materials and Methods

### 2.1. Patients

From January 2013 to February 2022, we retrospectively screened all patients older than 18 years who were admitted with a diagnosis of status epilepticus to the medical ICU of Lille University Hospital and requiring mechanical ventilation. Patients were screened to confirm the diagnosis of SE—meaning no other possible diagnosis could be considered—and the absence of exclusion criteria.

### 2.2. Inclusion and Exclusion Criteria

The selected patients had generalized convulsive SE, defined as 5 or more minutes of continuous clinical seizure activity or two seizures without a return to baseline in the interval [2], and had received MV.

Exclusion criteria included post-anoxic SE due to the heterogeneity of their management [8].

According to French law, this database was declared at the institutional data protection board (DEC19-432, DEC20-354), and the study was approved by our local ethics committee (CE SRLF 21-38). This research has therefore been performed in accordance with the ethical standards laid down in the 1964 Declaration of Helsinki and its later amendments. The authors have full access to all data and have the right to publish all data, separate and apart from the guidance of any sponsor.

### 2.3. Data Collection

For each patient, demographic characteristics and medical history were recorded. Severity at admission was defined using the Simplified Acute Physiology Score II (SAPS II) with exclusion of age studied separately: higher scores indicated greater severity of illness [9]. Clinical characteristics of SE were reported. Refractory status epilepticus was declared if the initial treatment failure included at least one benzodiazepine (i.e., clonazepam) and one intravenous long-duration antiepileptic drug (i.e., fos/phenytoin, levetiracetam, valproic acid, or phenobarbital) prior to intubation [10]. The etiology of SE was defined according to the international league against epilepsy (ILAE) classification: acute (e.g., stroke, intoxication, encephalitis), remote (e.g., poststroke, posttraumatic), progressive (e.g., brain tumor, dementias), and unknown [2]. Psychogenic non-epileptic seizure diagnosis was based on a paroxysmal event without ictal epileptiform EEG changes [11]. Due to a difficult diagnosis at ICU admission and a history of epilepsy frequently associated, these patients were not excluded. We also defined groups of etiology as vascular (acute SE related to ischemic or hemorrhagic stroke, cerebral venous thrombosis, and posterior cerebral encephalopathy), toxic (acute SE related to metabolic disturbance, alcohol, drug intoxication, or withdrawal), and brain tumor (progressive SE related to brain tumor).

Clinical, biological, radiological, and microbiological diagnostic criteria for BAP, as well as clinical outcomes (duration of MV, ICU length of stay, ICU mortality, 3-month mortality, and functional outcome), were collected. Functional outcome was evaluated by modified Rankin Scale (mRS) [12] during a face-to-face visit or by a telephone interview with the patient, the family, or the general practitioner. A poor functional outcome was defined as mRS score >1 and was different from the pre-SE mRS score.

### 2.4. Definitions

#### 2.4.1. Aspiration Syndrome

The diagnosis of aspiration syndrome was based on the presence of at least two of the following criteria during the first 2 days after initiation of MV: body temperature of more than 38.5 °C or less than 35.5 °C; leucocyte count greater than 12,000 cells per μL or less than 4000 cells per μL, and purulent tracheal secretions; and the presence of new or progressive infiltrates on the chest X-ray. Chest X-rays were reviewed by at least two attending physicians. In the case of disagreement, a third physician was asked to interpret the chest radiograph. Of note, we collected macroaspiration, defined by history of vomiting before or during intubation, which was not required to define aspiration syndrome.

#### 2.4.2. Bacterial Aspiration Pneumonia, Pneumonitis, and Ventilator-Associated Pneumonia

All aspiration syndromes were classified as [7]

Bacterial aspiration pneumonia in the case of microbiological confirmation, with the isolation in the endotracheal aspirate of at least 10^5^ colony-forming units per mL.Pneumonitis, when endotracheal aspirate culture was sterile.

Ventilator-associated pneumonia (VAP) had the same diagnostic criteria as BAP but occurred at least 2 days after starting MV.

#### 2.4.3. Measurements of Serum Levels of C-Reactive Protein (CRP), Procalcitonin (PCT), and Albumin during First 24 h after Admission

CRP was measured with the immunoturbidimetric method and a detection limit of 0.3 mg/L. PCT concentrations were determined using an electrochemiluminescence immunoassay with a detection limit of 0.02 ng/mL. Albumin concentrations were measured with the immunoturbidimetric method. All measurements were performed with Cobas 8000 modular analyzer series (Roche Diagnostics, Rotkreuz, Switzerland).

### 2.5. Outcomes

The primary outcome was the incidence of bacterial aspiration pneumonia, occurring during the first 2 days after starting invasive MV, among patients admitted to ICU with generalized convulsive SE requiring MV. The secondary outcomes were the incidence of aspiration syndrome in order to identify factors associated with BAP, as well as MV duration, ICU length of stay, functional outcome, and death at 3 months.

### 2.6. Statistical Analysis

Categorical variables were expressed as numbers and percentages and were compared with the use of ordinal chi-squared or Fisher’s exact tests, as appropriate. Continuous variables were expressed as medians (interquartile ranges) and were compared with the use of a t-test, Welch’s test, or Wilcoxon signed-rank test, as appropriate. A Shapiro–Wilk test was used to distinguish between normal and abnormal distributions.

Logistic multivariable analysis was performed for the occurrence of BAP. To avoid overfitting, only variables with *p*-values under 0.10 in the univariate analysis were considered for inclusion in the final model. Multicollinearity was assessed using variance inflation factor with a cut-off at 4. Clinical relevance of variables was discussed. The fitness was evaluated by Negelkerke’s R^2^.

All tests were two sided, and the statistical significance was defined by *p*-values under 0.05. Statistical analyses were performed with R statistical software, version 3.6.0 [13].

## 3. Results

From January 2013 to February 2022, 246 patients were screened for eligibility. Among them, 20 patients with postanoxic SE were excluded. Two hundred and twenty-six patients were admitted to ICU with generalized convulsive SE requiring MV (Figure 1).

### 3.1. Patient Characteristics

The median age was 55 years (interquartile range, 43 to 68), and 146 (65%) patients were males. A total of 109 (48%) patients had a history of epilepsy and 40 (18%) of previous SE. Fifty-five (24%) patients were considered as refractory status epilepticus. According to ILAE’s etiologic categories, 71 (31%) patients had a SE related to an acute brain injury, 104 (46%) to previous and stable brain lesion (remote symptomatic), 31 (14%) to a progressive brain injury, and 14 (6%) patients had an SE of unknown origin (Table 1).

### 3.2. Comparison of Patients with and without Aspiration Syndrome

One hundred and three patients (46%) met the criteria for aspiration syndrome. Of note, considering the overall cohort, these criteria were frequently found during the first 48 *h* of MV: 147 (66%) had abnormal body temperature, 160 (72%) had purulent tracheal aspirates, and 171 (77%) had leukocytosis.

At admission, patients with aspiration syndrome had more frequently persistent seizures (*p* = 0.001). Body temperature was similar (*p* = 0.90) and heart rate was higher in patients with aspiration syndrome (*p* = 0.01). No other baseline characteristics differed, especially SE characteristics and etiology. Interestingly, macroaspiration (*p* = 0.50) and timing (*p* = 0.53) or reason (*p* = 0.54) for intubation were not associated with aspiration syndrome (Table 1).

Regarding laboratory results, patients with aspiration syndrome had a lower Pa0_2_/Fi0_2_ ratio (*p* < 0.001), a higher serum level of CRP (*p* < 0.01), and a lower serum level of albumin (*p* = 0.03). No difference was found for serum blood level of PCT (*p* = 0.13) and leukocyte count (*p* = 0.54) (Table 1).

Aspiration syndrome was associated with a longer MV duration (*p* < 0.01) and ICU length of stay (*p* = 0.01). Three-month mortality (*p* = 0.43) and poor functional outcome (*p* = 0.36) did not differ between groups (Table 1).

### 3.3. Comparison of Patients with BAP Versus Pneumonitis

Among patients with aspiration syndrome, 12 (5%) did not have endotracheal aspirate and could not be classified as BAP or pneumonitis. These patients were excluded for analysis concerning BAP and pneumonitis (Figure 1). Fifty-four patients (59%) had a BAP, whereas others were considered as pneumonitis. Considering the overall cohort, 24% of patients with GCSE presented a BAP.

Patients with BAP, in comparison with pneumonitis, were less likely alcoholic (*p* = 0.02) and had a higher SAPS II at admission (*p* = 0.03). No other baseline characteristics differed, especially SE characteristics and etiology (Table 2).

We found no difference concerning serum levels of CRP (*p* = 0.78), PCT (*p* = 0.87), and albumin (*p* = 0.41) between patients with BAP and pneumonitis. The severity of hypoxia estimated by the Pa0_2_/Fi0_2_ ratio and arterial lactate level did not differ between groups (respectively, *p* = 0.34 and *p* = 0.31) (Table 2).

We did not perform a multivariable analysis due to the absence of clinical relevance.

### 3.4. Etiology of Bacterial Aspiration Pneumonia

Of the 54 patients with BAP, 71 different bacterial strains were identified. They are presented in Table 3 with their antimicrobial susceptibility. Fifteen (28%) patients had a polymicrobial BAP. The most represented bacteria were *Staphylococcus aureus* (18; 33%), *Haemophilus influenzae* (13; 24%), *Streptococcus pneumoniae* (10; 19%) and *Klebsiella pneumoniae* (9; 17%).

### 3.5. Antibiotic Therapy

A total of 158 patients (70%) were treated by antibiotic therapy: 138 (87%) for suspicion of BAP and 20 (13%) for documented or suspected infection other than BAP.

Among patients with suspicion of BAP, 45 patients (33%) did not have an aspiration syndrome and 120 (87%) were treated by ACA.

Among the 54 patients with BAP, 13 (24%) patients presented at least one bacterium with no ACA-susceptibility. These patients more frequently had a colonization with ESBL-producing bacteria (*p* = 0.01) and received more antibiotics during the 3 months before ICU admission (*p* < 0.01) (Appendix A Appendix A). Among them, initial antibiotic therapy was inappropriate in 11 patients (85%). None presented an acute respiratory syndrome distress, or a septic shock or death related to respiratory infection.

The median duration of treatment for BAP was 7 days (5–7). In comparison with BAP, patients with pneumonitis did not have a shorter duration of antibiotic therapy (*p* = 0.46).

### 3.6. Outcomes of BAP Versus Pneumonitis

Patients with BAP, as compared with those with pneumonitis, did not have a longer duration of MV (*p* = 0.18) and ICU stay (*p* = 0.18). Three-month mortality was 17% versus 8%, respectively, in patients with BAP and pneumonitis (*p* = 0.20). Twenty-four patients (44%) in the BAP group versus ten (27%) in pneumonitis group had a 3-month poor functional outcome (*p* = 0.06) (Table 4).

## 4. Discussion

In the present study, we found the following: (1) Half the patients with generalized convulsive SE requiring mechanical ventilation had an aspiration syndrome and one-quarter suffered from BAP. (2) No clinical and laboratory characteristics allowed for the separation of BAP from pneumonitis. In particular, biomarkers related to inflammatory response such as serum CRP, PCT, and albumin were not associated with BAP and rather were modified by aspiration syndrome, including pneumonitis. (3) Up to one-quarter of bacteria isolated from tracheal samples was not sensitive to ACA, with no clinical consequence (i.e., septic shock, acute respiratory distress syndrome). (4) Finally, we did not find that BAP was associated with a poor functional outcome and death at 3 months compared to pneumonitis.

Aspiration syndrome is a frequent condition occurring in cases of impairment of consciousness [14]. In comparison with patients requiring MV for coma (epilepsy was the etiology of coma in 14%), we found a higher frequency of aspiration syndrome and BAP [7]. This result could be related to a higher risk of aspiration during SE. Indeed, in our study, persistent convulsions at admission were associated with aspiration syndrome. In another study of patients requiring MV for coma (13% had a convulsive SE), the authors found a higher frequency of aspiration syndrome (81 patients, 79%) and more BAP (45 patients, 44%) than in our cohort [15].

As previously described, aspiration syndrome was responsible for a longer duration of MV and ICU stay, without impact on ICU mortality [7].

Differentiation BAP from pneumonitis is the keystone of antibiotic management in order to avoid useless prescriptions and the emergence of resistant bacteria. In line with a previous study, we found no relevant difference between BAP and pneumonitis, including previous use of a proton pump inhibitor [7,16]. Serum levels of acute-phase proteins of inflammation were not modified by BAP. PCT increases under various inflammatory conditions, especially in the case of bacterial infections [17,18]. Some studies tried to use PCT to distinguish pneumonitis from BAP with no significant difference [7,19]. Only one study found that serum levels of CRP and PCT increased in cases of BAP with a poor diagnostic value [15]. However, they compared BAP versus no BAP (including pneumonitis and absence of aspiration syndrome), and the results could be biased by the presence of patients without aspiration syndrome who do not have increased serum levels of CRP and PCT [15]. Interestingly, early elevation of PCT is rather associated with poor neurological outcome in acute brain injury, such as SE [20,21], post-cardiac arrest syndrome [22], and stroke [23]. Currently, only microbiological culture of the tracheal sample differentiates these two entities with a result obtained in 2–3 days [24]. One elegant method to rapidly differentiate pneumonitis from BAP is to use polymerase chain reaction (PCR)-based techniques [25,26]. The results could be obtained in a few hours, reducing antimicrobial consumption in the ICU and breaking the vicious circle of multidrug-resistant bacteria emergence [27]. Furthermore, in the case of SE, some experimental and clinical data suggest that antibiotic therapy could increase the risk of symptomatic seizures in combination with renal dysfunction, brain lesion, and epilepsy [28].

Microbiology of BAP with a predominance of *Staphylococcus aureus*, *Haemophilus influenzae*, and *Streptococcus pneumoniae* is in line with a previous study [29].

Infections in SE are already known as factors associated with poor outcome [5]. The impact on MV duration and ICU stay is uncertain, in contrast to aspiration syndrome, but could be increased in BAP [7,15]. Further, previous studies reported the absence of significant relationship between BAP and ICU mortality [7,15]. However, our study follow-up at 3 months highlighted a possible higher mortality and morbidity in BAP versus pneumonitis. A large prospective study is needed to confirm these results.

Antibiotic therapy was frequently initiated for suspicion of BAP. However, one-third of patients did not have aspiration syndrome. Two reasons could explain this early antibiotic treatment: the presence of macroaspiration [30] and the elevation of inflammatory biomarkers such as serum levels of CRP and PCT. We found that macroaspiration, defined by history of vomiting before or during intubation, was not predictive of aspiration syndrome; and modifications of acute-phase proteins of inflammation, such as CRP and PCT were associated with aspiration syndrome rather than BAP.

One-quarter of patients with BAP had at least one bacterium that was non-susceptible to ACA. ACA, as alternative to ceftriaxone, is recommended for treatment of respiratory infection occurring during the first 5 days after admission [31]. Antibiotics before ICU admission and colonization with ESBL-producing bacterium are well known factors associated with ACA resistance [31]. In another study of early onset VAP (2–5 days after mechanical ventilation starting, corresponding to ICU admission) in a French neuro-ICU, the authors found 36% of patients with at least one bacterium resistant to ACA, mainly in patients with antibiotics before admission [32]. These data are in line with previous studies [31]. It seems important to determine new specific factors associated with ACA resistance and to develop a PCR-based method to rapidly identify antimicrobial resistance.

The median duration of antibiotic therapy for suspicion of BAP was 7 days, in line with the actual recommendation [33]. However, in the case of pneumonitis, we did not show a shorter duration, probably due to the delay in obtaining the result of microbiological culture and the fear of false-negative results [30]. In one study of patients with coma requiring MV, discontinuation of antibiotic therapy when no microorganism was found did not increase morbidity or mortality and tended to decrease the time with antibiotics during first 8 days [7]. The excellent predictive value of PCR-based methods would be interesting to use early on to differentiate pneumonitis from BAP, in order to avoid antibiotic treatment or to allow early withdrawal in patients with negative results and so decrease antibiotic consumption. This strategy is being studied in an ongoing randomized controlled trial (NCT03763799).

Our study has some strengths. Patients were similar in their clinical presentation with only GCSE. Post-anoxic SE was excluded due to the heterogeneity of their management [8]. The monocentric design contributed to a homogeneous management, especially in SE treatment and BAP diagnosis. Baseline characteristics and the main results were in line with previous studies [7,15]. Infection was considered only when confirmed by culture.

Our study also has some limitations. Monocentric design contributes to the small size of the cohort and results should be applied only for generalized convulsive SE. However, our main results are similar to those of a study including different causes of coma [7]. The long duration of data collection could impact the results. However, the methodology to determine infection did not change during the study. The primary outcome was the incidence of bacterial aspiration pneumonia, and a potential modification of bacterial ecology could not affect this variable. We did not have a control group to compare incidence of BAP between patients with SE and patients with impaired consciousness not due to SE. No anaerobic microorganisms were identified by microbiological cultures. They may play a role in BAP; however, they are often mixed with aerobic bacteria [29].

## 5. Conclusions

In conclusion, among patients with generalized convulsive SE requiring MV, half had criteria for aspiration syndrome and one-quarter of patients had a BAP. Clinical characteristics and biomarkers, such as serum CRP, PCT, and albumin levels measured during the first 24 h, were not useful to differentiate BAP from pneumonitis. Up to one-quarter of bacteria isolated from tracheal samples were not sensitive to ACA, with no clinical consequences despite inappropriate antibiotic therapy. BAP tended to be associated with poor functional outcome and death at 3 months. Our results highlight the need for a method to rapidly differentiate pneumonitis from BAP, such as PCR-based techniques.

## Figures and Tables

**Figure 1 jcm-11-06673-f001:**
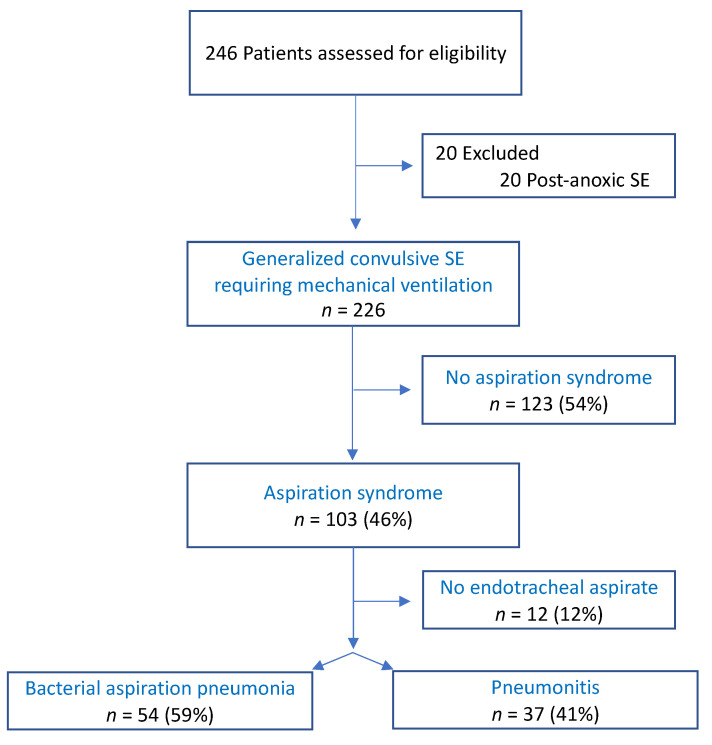
Flowchart. Abbreviations: SE, status epilepticus.

**Table 1 jcm-11-06673-t001:** Characteristics and outcomes of patients with and without aspiration syndrome.

Characteristics	Overall Cohort (*n* = 226)	Aspiration Syndrome (*n* = 103)	No Aspiration Syndrome (*n* = 123)	*p*-Value
**Demographics**				
Age (years), median (IQR)	55 (43–68)	56 (43–68)	55 (39–67)	0.395
Male sex, *n* (%)	146 (65)	66 (64)	80 (65)	0.880
mRS score 0–1, *n* (%)	132 (58)	53 (51)	79 (64)	0.052
SAPS II (without age), median (IQR)	46 (36–54)	50 (34–58)	45 (37–50)	0.113
**Medical history, *n* (%)**				
History of epilepsy	109 (48)	50 (49)	59 (48)	0.931
Previous status epilepticus	40 (18)	14 (14)	26 (21)	0.139
Alcohol	83 (37)	43 (42)	40 (33)	0.152
**Status epilepticus characteristics, *n* (%)**				
Continuous seizure	72 (32)	32 (31)	40 (33)	0.815
Refractory status epilepticus	55 (24)	22 (21)	33 (27)	0.340
Intubation before ICU admission	128 (57)	56 (54)	72 (59)	0.529
Intubation for respiration failure	130 (58)	57 (55)	73 (59)	0.544
**Clinical characteristics at admission**				
Core temperature (°C), median (IQR)	36.9 (36.2–37.4)	36.9 (36.2–37.4)	36.8 (36.3–37.4)	0.897
Heart rate (bpm), median (IQR)	97 (80–119)	102 (85–125)	95 (78–112)	**0.011**
Persistent seizures, *n* (%)	43 (19)	29 (28)	14 (11)	**0.001**
History of macroaspiration, *n* (%)	23 (10)	12 (12)	11 (9)	0.503
**Biological characteristics at admission, median (IQR)**				
Leukocyte count (10^9^/L)	11.62 (8.88–15.63)	12.12 (9.55–15.67)	11.10 (8.47–15.20)	0.538
Serum CRP level (mg/L)	10 (0–27)	15 (4–36)	6 (0–18)	**0.002**
Serum PCT level (ng/mL)	0.15 (0.00–0.54)	0.18 (0.00–0.91)	0.10 (0.00–0.30)	0.134
Serum albumin level (g/L)	36 (32–40)	35 (31–38)	36 (32–40)	**0.030**
PaO_2_/FiO_2_ (mmHg)	286 (200–390)	265 (179–335)	313 (223–410)	**<0.001**
Arterial lactate level (mmol/L)	2.1 (1.2–4.0)	2.0 (1.1–3.6)	2.2 (1.3–4.2)	0.569
**Etiologic categories, *n* (%)**				
Acute symptomatic	71 (31)	33 (32)	38 (31)	0.853
Remote symptomatic	104 (46)	46 (45)	58 (47)	0.708
Progressive symptomatic	31 (14)	13 (13)	18 (15)	0.661
Unknown	14 (6)	10 (10)	4 (3)	**0.045**
PNES	6 (3)	1 (1)	5 (4)	0.224 *
**Main etiologies, *n* (%)**				
Vascular	26 (11)	10 (10)	16 (13)	0.439
Toxic	34 (15)	20 (19)	14 (11)	0.092
Brain tumor	25 (11)	14 (14)	11 (9)	0.267
**Outcomes**				
Ventilator-associated pneumonia, *n* (%)	35 (15)	15 (15)	20 (16)	0.725
Mechanical ventilation duration (days), median (IQR)	1.90 (0.88–4.87)	2.71 (1.50–7.10)	1.50 (0.60–3.65)	**0.002**
ICU stay length (days), median (IQR)	4.94 (2.71–9.69)	6.04 (4.23–12.93)	3.90 (2.11–6.92)	**0.011**
ICU mortality, *n* (%)	11 (5)	7 (7)	4 (3)	0.217
Three-month poor functional outcome, *n* (%)	78 (35)	38 (38)	40 (33)	0.363
Three-month mortality, *n* (%)	25 (11)	13 (13)	12 (10)	0.429

Categorical variables were expressed as number (percentage) and compared by a chi-squared test or Fisher’s exact test when specified by *. Continuous variables were expressed as a median (inter-quartile range), and a *t*-test was performed (Welch or Wilcoxon tests, as appropriate). Missing values, overall cohort (aspiration syndrome; no aspiration syndrome): leukocyte count, 3 (0;3); serum CRP level, 20 (6;14); serum PCT level, 50 (19;31); serum albumin level, 21 (9;12); PaO_2_/FiO_2_, 1 (0;1); lactate arterial level, 3 (1;2); 3-month outcomes, 4 (4;0). Abbreviations: mRS, modified Rankin scale; SAPS II, simplified acute physiology score; ICU, intensive care unit; CRP, C-reactive protein; PCT, procalcitonin; PNES, psychogenic non-epileptic seizure.

**Table 2 jcm-11-06673-t002:** Comparison of patients’ characteristics according to the diagnosis of BAP or pneumonitis.

Characteristics	BAP (*n* = 54)	Pneumonitis (*n* = 37)	*p*-Value
**Demographics**			
Age (years), median (IQR)	55 (42–67)	65 (48–71)	0.126
Male sex, *n* (%)	37 (69)	25 (68)	0.924
mRS score 0–1, *n* (%)	26 (48)	21 (57)	0.420
SAPS II (without age), median (IQR)	51 (38–58)	44 (28–58)	**0.035**
**Medical history, *n* (%)**			
History of epilepsy	25 (46)	18 (49)	0.825
Previous status epilepticus	4 (7)	9 (24)	**0.023**
Alcohol	17 (31)	21 (57)	**0.016**
Use of proton pump inhibitor	19 (35)	13 (35)	0.996
**Status epilepticus characteristics, *n* (%)**			
Continuous seizure	17 (31)	11 (30)	0.859
Refractory status epilepticus	11 (20)	9 (24)	0.655
Intubation before ICU admission	29 (54)	21 (57)	0.774
Intubation for respiration failure	34 (63)	17 (46)	0.108
**Clinical characteristics at admission**			
Core temperature (°C), median (IQR)	36.8 (36.1–37.1)	37.0 (36.3–37.8)	0.095
Heart rate (bpm), median (IQR)	96 (81–120)	111 (90–124)	0.191
Persistent seizures, *n* (%)	15 (28)	10 (27)	0.937
History of macroaspiration, *n* (%)	4 (7)	6 (16)	0.306 *
**Biological characteristics at admission, median (IQR)**			
Leukocyte count (10^9^/L)	12.28 (9.58–15.31)	11.88 (9.53–15.68)	0.792
Serum CRP level (mg/L)	12 (2–36)	13 (0–31)	0.777
Serum PCT level (ng/mL)	0.17 (0.00–0.63)	0.14 (0.00–0.53)	0.869
Serum albumin level (g/L)	35 (32–39)	35 (31–37)	0.411
PaO_2_/FiO_2_ (mmHg)	281 (185–391)	263 (192–314)	0.344
Arterial lactate level (mmol/L)	1.85 (1.00–3.17)	2.40 (1.42–4.92)	0.306
**Etiologic categories, *n* (%)**			
Acute symptomatic	15 (28)	13 (35)	0.455
Remote symptomatic	24 (44)	16 (43)	0.910
Progressive symptomatic	8 (15)	4 (11)	0.755 *
Unknown	6 (11)	4 (11)	0.964 *
**Main etiologies, *n* (%)**			
Vascular	5 (9)	2 (5)	0.696 *
Toxic	9 (17)	9 (24)	0.368
Brain tumor	8 (15)	5 (14)	0.862

Categorical variables were expressed as number (percentage) and compared by a chi-squared test or Fisher’s exact test when specified by *. Continuous variables were expressed as median (inter-quartile range), and a *t*-test was performed (Welch or Wilcoxon tests, as appropriate). Missing values (BAP; pneumonitis): CRP, 6 (5;1); PCT, 19 (12;7); albumin, 7 (3;4); lactate arterial, 1 (0;1). Abbreviations: BAP, bacterial aspiration pneumonia; mRS, modified Rankin scale; SAPS II, simplified acute physiology score; ICU, intensive care unit; CRP, C-reactive protein; PCT, procalcitonin.

**Table 3 jcm-11-06673-t003:** Bacterial strains identified by endotracheal aspirate in patients with bacterial aspiration pneumonia.

Type of Bacteria	Bacteria	Number of Isolates, *n* (%)	Resistance
**Gram +** (30 bacteria isolated from 29 patients, 54 %)			
***Staphylococcus* spp.**	*Staphylococcus aureus*	18 (33)	Methicillin-sensitive: 16 (penicillin-resistant: 8, tested in 10 isolates) Methicillin-resistant: 2 *
***Streptococcus* spp.**	*Streptococcus pneumoniae*	10 (19)	Wild-type: 6 Decreased susceptibility to penicillin: 4
	*Streptococcus agalactiae*	2 (4)	Wild-type: 2
**Gram −** (41 bacteria isolated from 36 patients, 67 %)			
**Group 1, 2, and 5 enterobacterales**	*Klebsiella pneumoniae*	9 (17)	Wild-type: 6 β-Lactamase: 1 * ESBL: 2 *
	*Escherichia coli*	7 (13)	Wild-type: 3 Low-production of β-lactamase: 1 Hyperproduction of β-lactamase: 3 *
	*Klebsiella oxytoca*	1 (2)	Wild-type: 1
	*Proteus vulgaris*	1 (2)	Wild-type: 1
**Group 3** **enterobacterales**	*Enterobacter cloacae*	2 (4)	Wild-type: 1 * ESBL: 1 *
	*Hafnia alvei*	1 (2)	Wild-type: 1 *
**Non-fermenting bacilli**	*Pseudomonas aeruginosa*	2 (4)	Wild-type: 2 *
	*Acinetobacter baumannii*	1 (2)	Wild-type: 1 *
**Other bacteria**	*Haemophilus influenzae*	13 (24)	Wild-type: 12 β-Lactamase: 1
	*Moraxella catarrhalis*	2 (4)	β-Lactamase: 1 Wild-type: 1
	*Lelliottia amnigena*	1 (2)	ACA-resistant: 1 *
	*Pasteurella multocida*	1 (2)	Wild-type: 1

Of the 54 patients with BAP, 71 different bacterial strains were identified. Antimicrobial susceptibility was defined, and * corresponds to ACA-resistant bacteria. Abbreviations: ESBL, extended spectrum beta-lactamase; ACA, amoxicillin-clavulanic acid.

**Table 4 jcm-11-06673-t004:** Comparison of patients’ outcomes according to the diagnosis of BAP or pneumonitis.

Characteristics	BAP (*n* = 54)	Pneumonitis (*n* = 37)	*p*-Value
Ventilator-associated pneumonia, *n* (%)	10 (19)	4 (11)	0.317
Mechanical ventilation duration (days), median (IQR)	3.39 (1.71–8.35)	2.37 (0.83–6.88)	0.179
ICU stay length (days), median (IQR)	6.83 (4.38–13.25)	6.46 (3.58–12.50)	0.180
ICU mortality, *n* (%)	6 (11)	1 (3)	0.234 *
Three-month poor functional outcome, *n* (%)	24 (44)	10 (27)	0.057
Three-month mortality, *n* (%)	9 (17)	3 (8)	0.198

Categorical variables were expressed as number (percentage) and compared by a chi-squared test or Fisher’s exact test when specified by *. Continuous variables were expressed as median (inter-quartile range), and a t-test was performed (Welch or Wilcoxon tests, as appropriate). Missing values (BAP; pneumonitis): 3-month outcomes, 3 (3;0). Abbreviations: BAP, bacterial aspiration pneumonia; ICU, intensive care unit.

## Data Availability

The data presented in this study are available on request from the corresponding author. The data are not publicly available due to privacy reasons.

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
