# Peer review of "Bacterial Aspiration Pneumonia in Generalized Convulsive Status Epilepticus: Incidence, Associated Factors and Outcome"

_jcm, 2022, doi:10.3390/jcm11226673_

Round 1

Reviewer 1 Report

Nice to read a draft of this kind which is focused, documented and very pragmatic. 

An idea for the future is a subset analysis according to bacterial strains (ie which are the most clinically burdensome among the strains  eg which strain is more likely to be associated with ARDS etc) 

Otherwise "chapeau"

Author Response

            Thank you for your comment. Concerning the subset analysis, the complications of bacterial aspiration pneumonia (e.g., ARDS or septic shock) are uncommon. In our study, none presented an acute respiratory syndrome distress, or a septic shock or death related to respiratory infection. However, a large prospective study is needed to confirm our results and focus on bacterial resistances and outcomes.

Reviewer 2 Report

I am glad to review your manuscript.

1) As the authors suggested in the limitation, no anaerobe was identified in microbiologic cultures, so I am wondering about the role of the anaerobe in the pneumonitis group.

2) Even though p-values defined the statistical significance under 0.05, many results quoted a tendency (lines 20, 177, 200, 240, 250). The tendency must be used carefully because of containing a positive response.

3) There are some errors in the denominator, please check up on lines 18 and 21.

4) Because of the retrospectively designed study, a large prospective study will be needed to confirm the results.

Thank you!

Author Response

Referee: 2

I am glad to review your manuscript.

1) As the authors suggested in the limitation, no anaerobe was identified in microbiologic cultures, so I am wondering about the role of the anaerobe in the pneumonitis group.

            The role of anaerobic bacteria in pulmonary infections is unclear. Concerning bacteria aspiration pneumonia (BAP), as discussed in manuscript, is often mixed with aerobic bacteria [1], and do not modify the classification into BAP. In this study, anaerobic bacteria were isolated in a minority of cases, despite the information of microbiologist and immediately transferred samples to microbiology unit [1]. Only 6 patients (5%) had a tracheal culture positive to anaerobic bacteria, including 5 patients with a co-infection to aerobic bacteria (S. aureus) [1]. Anaerobic bacteria are more likely associated with complicated pulmonary infections such as abscess, or pleuropneumonia [2]. Finally, the antimicrobial coverage of these bacteria is still debated: most of these bacteria have a susceptibility to cephalosporin or amoxicillin-clavulanate and metronidazole (or clindamycin) may have potential adverse effects [1].

            Your comment focuses on pneumonitis group. We cannot exclude a possible infection due to anaerobic bacteria. PCR-based techniques could help their detection. However, the pathogenic role of these bacteria in BAP is unclear, especially in absence of complications such as pulmonary abscess. We did not find data in literature concerning pneumonitis and anaerobic bacteria. The impact of these bacteria seems to be very limited.

2) Even though p-values defined the statistical significance under 0.05, many results quoted a tendency (lines 20, 177, 200, 240, 250). The tendency must be used carefully because of containing a positive response.

            We have changed these sentences to remove the term “tended”.

3) There are some errors in the denominator, please check up on lines 18 and 21.

            Thanks for noticing it! We also made changes in the table 4 and related paragraph.

4) Because of the retrospectively designed study, a large prospective study will be needed to confirm the results.

            We agree with this comment.

Thank you!

References

  1. Lauterbach, E.; Voss, F.; Gerigk, R.; Lauterbach, M. Bacteriology of Aspiration Pneumonia in Patients with Acute Coma. Intern Emerg Med 2014, 9, 879–885.
  2. DiBardino, D.M.; Wunderink, R.G. Aspiration Pneumonia: A Review of Modern Trends. J Crit Care 2015, 30, 40–48.

Reviewer 3 Report

Summary of article

Dr. Tortuyaux et al. investigated bacterial aspiration pneumonia (BAP) incidence and risk factors in patients with generalized convulsive status epilepticus (GCSE). Conducting retrospective medical record review, they included 226 patients over 18 years old with GCSE requiring mechanical ventilation. Comparing the patients with and without aspiration syndrome, the authors found that an aspiration syndrome group had higher heart rate, persistent seizures, higher serum CRP level, lower serum albumin level, and lower PaO2/FiO2 at admission, unknown etiology. The aspiration syndrome group had outcomes of longer mechanical ventilation duration and ICU stay length. Comparing the patients with BAP and pneumonitis, the authors found that the BAP group had worse SAPS II, a lower proportion of previous status epilepticus, and a lower proportion of medical history of alcoholism. There is statistical insignificance of all measured outcomes of patients with BAP compared to those with pneumonitis. The authors concluded that one-quarter of patients had a BAP among patients with GCSE. They also concluded that clinical characteristics and biomarkers were not useful in differentiating BAP from pneumonitis.

Comments (Invitation on Oct 6, 2022, and comment submission on Oct 8, 2022)

This study addressed an interesting topic of bacterial aspiration for patients with GCSE requiring mechanical ventilation. However, I have some concerns for the publication of this manuscript. Please consider addressing some concerns, as shown below.

Here are my comments and suggestions about this manuscript.

Major points:

[1] “Abstract”

The authors should not describe results as “longer duration” or “higher rate” when statistically insignificant.  

[2] “Introduction”

Please rewrite the hypothesis or aim of this study. The authors’ current hypothesis does not cover all the motivations for conducting this study. Specifically, the current hypothesis does not explain why the authors compared clinical characteristics between patients with and without aspiration syndrome. The motivation for identifying risk factors for BAP also does not cover my current hypothesis.

[3] “Methods”

The authors described “bivariate analysis” for the screening for the final multivariable logistic regression analysis. I think that should be described as “univariate analysis,” not “bivariate analysis.”

[4] “Methods”

Please clarify why the authors exclude patients over 18 years old.

[5] “Results”

The reviewer cannot understand why the authors did not perform a multivariate analysis to identify the risk factors for BAP because Table 2 showed some statistical significance between the BAP and Pneumonitis groups.

[6] “Method”

Please describe more details about PNES as an etiology of GCSE. The reviewer does not agree to include PNES as an etiology of GCSE. A reviewer recommends excluding the patients whose seizures' etiology was PNES.

Minor points:

[7] “Tables”

It is better to clarify the meaning of the numbers in parentheses (range, IQR, mean, or median?).

Author Response

Referee: 3

Dr. Tortuyaux et al. investigated bacterial aspiration pneumonia (BAP) incidence and risk factors in patients with generalized convulsive status epilepticus (GCSE). Conducting retrospective medical record review, they included 226 patients over 18 years old with GCSE requiring mechanical ventilation. Comparing the patients with and without aspiration syndrome, the authors found that an aspiration syndrome group had higher heart rate, persistent seizures, higher serum CRP level, lower serum albumin level, and lower PaO2/FiO2 at admission, unknown etiology. The aspiration syndrome group had outcomes of longer mechanical ventilation duration and ICU stay length. Comparing the patients with BAP and pneumonitis, the authors found that the BAP group had worse SAPS II, a lower proportion of previous status epilepticus, and a lower proportion of medical history of alcoholism. There is statistical insignificance of all measured outcomes of patients with BAP compared to those with pneumonitis. The authors concluded that one-quarter of patients had a BAP among patients with GCSE. They also concluded that clinical characteristics and biomarkers were not useful in differentiating BAP from pneumonitis.

This study addressed an interesting topic of bacterial aspiration for patients with GCSE requiring mechanical ventilation. However, I have some concerns for the publication of this manuscript. Please consider addressing some concerns, as shown below.

Major points:

[1] “Abstract”

The authors should not describe results as “longer duration” or “higher rate” when statistically insignificant. 

            We have modified these points.

[2] “Introduction”

Please rewrite the hypothesis or aim of this study. The authors’ current hypothesis does not cover all the motivations for conducting this study. Specifically, the current hypothesis does not explain why the authors compared clinical characteristics between patients with and without aspiration syndrome. The motivation for identifying risk factors for BAP also does not cover my current hypothesis.

            To describe the patients with and without aspiration syndrome is needed to better understand BAP. In the two main studies in this field, the authors described aspiration syndrome [1,2]. Nevertheless, some biological characteristics commonly related to infection (CRP and PCT), are rather associated with aspiration syndrome than BAP. To clarify this point, we now specify in the introduction that we aimed to described aspiration syndrome and BAP.

[3] “Methods”

The authors described “bivariate analysis” for the screening for the final multivariable logistic regression analysis. I think that should be described as “univariate analysis,” not “bivariate analysis.”

            We replace the word “bivariate” by “univariate” in the manuscript (line 141).

[4] “Methods”

Please clarify why the authors exclude patients over 18 years old.

            Patients over 18 years old were not exclude. We think the reviewer wonders about the exclusion of patients under 18 years old. We choose to exclude these patients to have a homogenous cohort of epileptic patients. Etiologies differ between patients under 18 years old (idiopathic and unknown etiologies) versus adults (mostly symptomatic etiologies). Moreover, the treatments are different and could impact length of mechanical ventilation, stay etc.

We propose to remove this exclusion criterion because no patient was excluded. Indeed, only patients older than 18 years were screened.

[5] “Results”

The reviewer cannot understand why the authors did not perform a multivariate analysis to identify the risk factors for BAP because Table 2 showed some statistical significance between the BAP and Pneumonitis groups.

            As specified in the manuscript, we did not perform a multivariable analysis due to absence of clinical relevance. To answer to your comment, we performed a multivariable analysis, using methodology described in the manuscript (method section). The variables included were SAPS II (age excluded), previous status epilepticus, medical history of alcoholism and core temperature. No variable was excluded due to multicollinearity. After exclusion of medical history of alcoholism by stepwise approach, we found that a higher SAPS II (age excluded) was associated with BAP (OR 1.02, 95%CI [1.00-1.05]; p=0.049). Patients with an history of SE had a lower risk of BAP (OR 0.33, 95%CI [0.11-0.99]; p=0.048). Core temperature at admission was not associated with BAP (OR 0.78, 95%CI [0.58-1.05]; p=0.110). The value of Nagelkerke’s R2 was 0.086. In our opinion, this analysis did not improve the manuscript.

[6] “Method”

Please describe more details about PNES as an etiology of GCSE. The reviewer does not agree to include PNES as an etiology of GCSE. A reviewer recommends excluding the patients whose seizures' etiology was PNES.

            The diagnosis of PNES is very difficult. Moreover, most of patients presenting PNES had also a diagnosis of epilepsy. Recent studies, including RCT (10% of the enrolled patients), did not exclude these patients [3]. A recent study about PNES in ICU found these patients had also a risk of ICU-related infection (16%) [4]. It is important to underline that diagnosis of PNES during ICU admission is very difficult and some patients had also interictal epileptic activities, mainly related to an history of epilepsy [4]. In our opinion, including these patients allows us to get closer to the practice.

So, we do not agree to exclude these patients. We now specify this point in the method section.

Minor points:

[7] “Tables”

It is better to clarify the meaning of the numbers in parentheses (range, IQR, mean, or median?).

            The meaning of the numbers in parentheses is specified in the text legend (and obviously in the method section). This is now specified in the tables.

References

  1. Lascarrou, J.B.; Lissonde, F.; le Thuaut, A.; Bachoumas, K.; Colin, G.; Henry Lagarrigue, M.; Vinatier, I.; Fiancette, M.; Lacherade, J.C.; Yehia, A.; et al. Antibiotic Therapy in Comatose Mechanically Ventilated Patients Following Aspiration: Differentiating Pneumonia From Pneumonitis. Crit Care Med 2017, 45, 1268–1275.
  2. Legriel, S.; Grigoresco, B.; Martel, P.; Henry-Lagarrigue, M.; Lvovschi, V.; Troché, G.; Amara, M.; Jacq, G.; Bruneel, F.; Bernard, M.; et al. Diagnostic Accuracy of Procalcitonin for Early Aspiration Pneumonia in Critically Ill Patients with Coma: A Prospective Study. Neurocrit Care 2019, 30, 440–448.
  3. Kapur, J.; Elm, J.; Chamberlain, J.M.; Barsan, W.; Cloyd, J.; Lowenstein, D.; Shinnar, S.; Conwit, R.; Meinzer, C.; Cock, H.; et al. Randomized Trial of Three Anticonvulsant Medications for Status Epilepticus. N Engl J Med 2019, 381, 2103–2113.
  4. Mezouar, N.; Demeret, S.; Rotge, J.Y.; Dupont, S.; Navarro, V. Psychogenic Non-Epileptic Seizure-Status in Patients Admitted to the Intensive Care Unit. Eur J Neurol 2021, 28, 2775–2779.

Round 2

Reviewer 3 Report

I appreciate the authors taking my suggestions and incorporating them into the manuscript.

I would like to congratulate their effort in the revision.